# Toward Tumor Fight and Tumor Microenvironment Remodeling: PBA Induces Cell Cycle Arrest and Reduces Tumor Hybrid Cells’ Pluripotency in Bladder Cancer

**DOI:** 10.3390/cancers14020287

**Published:** 2022-01-07

**Authors:** Carolina Rubio, José Avendaño-Ortiz, Raquel Ruiz-Palomares, Viktoriya Karaivanova, Omaira Alberquilla, Rebeca Sánchez-Domínguez, José Carlos Casalvilla-Dueñas, Karla Montalbán-Hernández, Iris Lodewijk, Marta Rodríguez-Izquierdo, Ester Munera-Maravilla, Sandra P. Nunes, Cristian Suárez-Cabrera, Miriam Pérez-Crespo, Víctor G. Martínez, Lucía Morales, Mercedes Pérez-Escavy, Miguel Alonso-Sánchez, Roberto Lozano-Rodríguez, Francisco J. Cueto, Luis A. Aguirre, Félix Guerrero-Ramos, Jesús M. Paramio, Eduardo López-Collazo, Marta Dueñas

**Affiliations:** 1Biomedical Biomedical Research Institute I+12, University Hospital “12 de Octubre”, Av Córdoba s/n, 28041 Madrid, Spain; carolina.rubio@externos.ciemat.es (C.R.); raquelruizpalomares@gmail.com (R.R.-P.); irisadriana.lodewijk@externos.ciemat.es (I.L.); ester.munera@ciemat.es (E.M.-M.); SandraIsabel.Pinto@externos.ciemat.es (S.P.N.); Cristian.Suarez@ciemat.es (C.S.-C.); Miriam.Perez@ciemat.es (M.P.-C.); VictorManuel.Garcia@ciemat.es (V.G.M.); MariaLucia@externos.ciemat.es (L.M.); MariaMercedes.Perez@ciemat.es (M.P.-E.); Miguel.alonso@ciemat.es (M.A.-S.); jesusm.paramio@ciemat.es (J.M.P.); 2Centro de Investigación Biomédica en Red de Cáncer (CIBERONC), 28029 Madrid, Spain; 3Molecular Oncology Unit, CIEMAT (Centro de Investigaciones Energéticas, Medioambientales y Tecnológicas), Avenida Complutense nº40, 28040 Madrid, Spain; Vikkaloy@gmail.com; 4TumorImmunology Laboratory and Innate Immunity Group, Institute for Health Research (IdiPAZ), 28029 Madrid, Spain; jose.avendano@idipaz.es (J.A.-O.); josecarlos.casalvilla.duenas@idipaz.es (J.C.C.-D.); Karlamarina.hernandez@idipaz.es (K.M.-H.); roberto.lozano.rodriguez@idipaz.es (R.L.-R.); aelopezc@salud.madrid.org (F.J.C.); luis.augusto.aguirre@idipaz.es (L.A.A.); 5Division of Hematopoietic Innovative Therapies, Centro de Investigaciones Energéticas, Medioambientales y Tecnológicas (CIEMAT), 28029 Madrid, Spain; Omaira.Alberquilla@ciemat.es (O.A.); Rebeca.Sanchez@externos.ciemat.es (R.S.-D.); 6Centro de Investigación Biomédica en Red de Enfermedades Raras (CIBER-ER), 28029 Madrid, Spain; 7Advanced Therapy Unit, Instituto de Investigación Sanitaria Fundación Jiménez Díaz (IIS-FJD/UAM), 28040 Madrid, Spain; 8Uro-Oncology Unit, 12 de Octubre University Hospital, Av Córdoba s/n, 28041 Madrid, Spain; martarodri86@gmail.com (M.R.-I.); felix.guerrero@salud.madrid.org (F.G.-R.); 9Cancer Biology and Epigenetics Group, Research Center of IPO Porto (CI-IPOP)/RISE@CI-IPOP (Health Research Network) Porto Comprehensive Cancer Center (Porto.CCC), 4200-072 Porto, Portugal; 10CIBER of Respiratory Diseases (CIBERES), 28029 Madrid, Spain

**Keywords:** bladder cancer, tumor microenvironment, tumor hybrid cells, HDACi

## Abstract

**Simple Summary:**

Bladder cancer (BC) is the second most frequent cancer of the genitourinary system. More than 500,000 patients per year are diagnosed with BC, a disease which additionally results in more than 200,000 annual deaths. One of the major problems in BC treatment is that many patients cannot receive appropriate treatment due to comorbidities and the severe inflammatory side effects of therapy. The aim of our study was to assess the effect of butyrate derivatives, demonstrating that they could be beneficial for treating the tumor and also to modify the tumor microenvironment. Upon treatment with butyrate derivatives, we particularly saw increased PD-L1 surface expression and reduced pluripotency molecular markers in a hybrid BC–macrophage cell population. This is a cell population known to display an increased capacity to migrate and evade immunity. Treatment with butyrate derivatives may also provide a better chance of immunotherapy success for BC patients.

**Abstract:**

Bladder cancer (BC) is the second most frequent cancer of the genitourinary system. The most successful therapy since the 1970s has consisted of intravesical instillations of *Bacillus Calmette–Guérin* (BCG) in which the tumor microenvironment (TME), including macrophages, plays an important role. However, some patients cannot be treated with this therapy due to comorbidities and severe inflammatory side effects. The overexpression of histone deacetylases (HDACs) in BC has been correlated with macrophage polarization together with higher tumor grades and poor prognosis. Herein we demonstrated that phenylbutyrate acid (PBA), a HDAC inhibitor, acts as an antitumoral compound and immunomodulator. In BC cell lines, PBA induced significant cell cycle arrest in G1, reduced stemness markers and increased *PD-L1* expression with a corresponding reduction in histone 3 and 4 acetylation patterns. Concerning its role as an immunomodulator, we found that PBA reduced macrophage IL-6 and IL-10 production as well as CD14 downregulation and the upregulation of both PD-L1 and IL-1β. Along this line, PBA showed a reduction in IL-4-induced M2 polarization in human macrophages. In co-cultures of BC cell lines with human macrophages, a double-positive myeloid–tumoral hybrid population (CD11b^+^EPCAM^+^) was detected after 48 h, which indicates BC cell–macrophage fusions known as tumor hybrid cells (THC). These THC were characterized by high PD-L1 and stemness markers (*SOX2*, *NANOG*, *miR-302*) as compared with non-fused (CD11b^−^EPCAM^+^) cancer cells. Eventually, PBA reduced stemness markers along with *BMP4* and *IL-10*. Our data indicate that PBA could have beneficial properties for BC management, affecting not only tumor cells but also the TME.

## 1. Introduction

Bladder cancer (BC) represents the ninth most frequent cancer worldwide and the second most frequent cancer of the genitourinary system, with more than 500,000 newly diagnosed cases per year and more than 200,000 annual deaths [1]. Approximately 75% of tumors arise from the urothelium in the form of non-muscle invasive BC (NMIBC; stages Ta, T1), whereas 25% display muscle-invasive disease (MIBC; stages ≥ T2) at diagnosis [2]. Since 1976, the most successful immunotherapy for NMIBC has been serial instillations of *Bacillus Calmette–Guérin* (BCG) [3], resulting in an activation of the immune response and the generation of an antitumor stroma that diminishes the recurrence and progression rates in responder patients. However, many patients cannot be treated with this therapy due to comorbidities and severe inflammatory side effects. Part of this tumor–stroma interaction involves both tumor-associated macrophages (TAM) and tumor-infiltrating lymphocytes (TIL) that favor the antigen presentation and elicit antitumor immune surveillance [4]. In addition, several studies have demonstrated that the pre-BCG baseline status of Th1/Th2 balance, regulatory T cell (Treg) recruitment and TAM polarization in the tumor microenvironment (TME) could influence the clinical response to BCG [5]. However, the knowledge of the impact of TAMs on tumor behavior and progression is rather limited.

It has been suggested that macrophages and malignant cells could form hybrid cells in vitro and in vivo with high metastatic potential [6,7,8]. The most accepted mechanism to explain how cancer cells undergo a phenotypic shift acquiring macrophage characteristics is through macrophage–tumor cell fusions. These macrophage–tumor cell hybrids display increased capacities to evade immunity and migrate [7,8,9]. We and others have demonstrated that tumor stem cells and monocytes/macrophages with an M2 phenotype are those more prone to generate tumor hybrid cells (THC) [6,10]. The importance of the cellular interaction between tumor cells and TAMs in BC has been substantiated by the observation of dense tumor cells with strong immunoreactivity with the endocytic receptor for haptoglobin–hemoglobin complexes (CD163, which is characteristic of M2 polarized macrophages) in BC tumor biopsies [11]. M2-like macrophage polarization is associated with higher tumor grade and worse prognosis in BC [12,13]. Specifically, conditioned media followed by the addition of tumor cell supernatant favors an M2-like polarization of macrophages through BMP4-dependent signaling. Indeed, BMP4 secretion by BC cells provides the M2 signal necessary for a pro-tumoral immune environment, which translates into cytokine production favoring tumor progression [12].

Several examples in the scientific literature suggest that the mechanism for the maintenance and fine-tuning of the BMP signaling pathway in the TME involves different regulatory loops, including protein–miRNAs interactions and epigenetic remodeling of the essential players. Kang et al. [14] showed a regulatory loop between BMP4–miR-302–BMPRII and suggested that the BMP4–miR-302 interaction was repressed by sodium butyrate, an HDAC inhibitor (HDACi). HDACis were initially studied for their anti-proliferative activity and are currently under evaluation for the treatment of solid tumors and hematological malignancies [15,16,17,18,19,20]. Following promising results, HDACis such as belinostat, panobinostat and vorinostat have been already approved for the treatment of some hematological malignancies. Recently, it was also shown that HDACis influence monocyte and macrophage differentiation, survival and function [21,22,23,24]. In particular, butyrate derivatives reduced inflammation in a mouse model of colitis [25], modifying macrophage polarization [26,27] and acting as neuroprotectors in a rat model of ischemia-induced brain injury [28]. Therefore, further evaluation of butyrate derivatives such as HDACis, not only as anti-tumor compounds but also as immunomodulators of the TME to condition an effective therapeutic effect, is needed.

In the present work, we evaluate the effect of butyrate derivatives (4-phenylbutyrate (PBA) and sodium phenyl butyrate (SB)) known for inducing differentiation, cell cycle arrest, and apoptosis in various cancer cells. HDACis has been demonstrated to be a pan-HDAC inhibitor acting over class I and class II HDACs [29]. We evaluate both compounds in BC cell lines, monocyte/macrophage polarization and in co-culture experiments. We observed that butyrate acts as an antitumor drug but can also modify the TME by affecting the surrounding immune cells and tumor–stroma interaction. Using co-cultures of both cell types, a THC population was obtained, which was shown to be more sensitive to butyrate treatment.

## 2. Materials and Methods

### 2.1. Cell Lines and Reagents

BC cell lines (J82, 5637, UMUC1), with known genomic characteristics [30] and validated by short-tandem repeat typing, were maintained in DMEM (Dulbecco’s Modified Eagle Medium GlutaMAX™) (Gibco-BRL Life Technologies, Grand Island, NY, USA) supplemented with 10% FBS (fetal bovine serum) and 1% antibiotic–antimycotic (Gibco-BRL Life Technologies) at 37 °C in a humidified atmosphere of 5% CO_2_. All cells were routinely checked for mycoplasma. PBA and SB (Sigma-Aldrich, Sigma-Aldrich, St. Louis, MO, USA) were dissolved in DMSO (Sigma-Aldrich) at a stock concentration of 2 M.

### 2.2. Cell Viability Assays

Cell proliferation was analyzed with XTT Cell Proliferation Kit II (Roche, Indianapolis, IN, USA). For the assay, cells were seeded at a density of 5000 cells per well in 96-well plates in six replicates. Before adding the compounds, cells were allowed to attach to the bottom of the wells for 24 h. After that, cells were treated with a range of concentrations of PBA and/or SB for 72 h. Cell viability was evaluated by the colorimetric assay XTT Cell Proliferation Kit II (Roche) and absorbance was measured at 490 nm using a Genios pro microplate reader (Tecan, Tecan Trading AG, Männedorf, Switzerland) in a 96-well plate. DMSO (0.1%) was used as a vehicle control. Fresh working solutions were prepared for immediate use. The half maximal inhibitory concentration (IC50) values of the different compounds were determined using Prism 9 (GraphPad Software, San Diego, CA, USA). Each experiment was performed three to five times.

### 2.3. Cell Cycle Analysis

Cells were seeded at a density of 30,000 cells/well in 12-well plates (Roche) and PBA or SB were added at a concentration ranging from 2 to 10 mM. A total of 72 h after treatment, cells were harvested, washed with phosphate-buffered saline (PBS) (Gibco-BRL Life Technologies) and fixed in 70% ice-cold ethanol overnight at 4 °C. Samples were then centrifuged and incubated in PBS containing 2 μg/mL DAPI (Roche) and 0.05% NP40 (Roche) for at least 2 h. Finally, cells were analyzed by flow cytometry using Becton Dickinson LSR Fortessa cell analyzer (BD Biosciences, San Jose, CA, USA)and BD FACSDiva software (BD Biosciences, San Jose, CA, USA). Data analysis was performed with FlowJo 7.6.5 software (BD, Ashland, OR, USA) using the cell cycle analysis tool. Each experiment was carried out at least 4 times, with 3 technical replicates for each condition per experiment.

### 2.4. Western Blot and Immunohistochemistry

Cells were disrupted in a lysis buffer (20 mM HEPES pH 7.5, 1% Triton X-100, 40 mM β-glycerophosphate, 100 mM NaCl, 20 mM MgCl_2_, 10 mM EGTA) supplemented with protease (complete Mini Protease Inhibitor Cocktail, Roche) and phosphatase inhibitor (PhosSTOP, Roche) cocktails and centrifuged to obtain the supernatant containing total protein. Per sample, 40 µg of total protein was resolved in 4–12% NuPAGE gels (Invitrogen, Carlsbad, CA, USA) and transferred to nitrocellulose membranes. Membranes were blocked with PBS containing 5% non-fat dry milk and 0.1% TWEEN 20 and incubated with the appropriate antibody dilution. Primary and secondary antibodies were diluted in blocking solution (Appendix A). Secondary antibodies were purchased from Jackson ImmunoResearch (Ely, UK). Super Signal West Pico Chemiluminescence Substrate (Thermo Scientific, Waltham, MA, USA) was used according to the manufacturer’s recommendations to visualize the bands.

### 2.5. Quantitative PCR with Reverse Transcription (RT-qPCR)

Gene expression was analyzed by RT-qPCR in J82, 5637 and UMUC1 cell lines after 72 h of PBA treatment. Total RNA was isolated using miRNeasy Mini Kit (QIAGEN GmbH, Hilden, Germany) according to the manufacturer’s instructions, with a final DNAse treatment (RnaseRNAse-Free Dnase Set; QIAGEN). Reverse transcription was performed using the Omniscript RT Kit (Qiagen) and a primer mix specific for all genes of interest and random primers (Sigma Aldrich) using 10 ng of total RNA. PCR was performed in a 7500 Fast Real-Time PCR System (Applied Biosystems, Waltham, MA, USA) using GoTaq PCR master mix (Promega, Madison, WI, USA) and 1 μL of cDNA as a template. Melting curves were performed to verify the specificity and absence of primer dimerization. Reaction efficiency was calculated for each primer combination. *GUSB* and *TBP* genes were used as housekeeping genes for normalization. The sequences of the specific oligonucleotides used are listed in Appendix A.

To measure miRNA expression quantitatively, RNA was extracted using the method mentioned above. Reverse transcription was carried out from 10 ng of total RNA along with miR-specific primers using the TaqMan MicroRNA Reverse Transcription Kit (Applied Biosystems). PCR assays were performed using TaqMan Gene Expression Master Mix and 7500 Fast Real Time PCR System (Applied Biosystems). *RNU6B* was used as a housekeeping gene.

### 2.6. Human Macrophage Isolation and Co-Cultures

Healthy donors were recruited from the Blood Donor Services of La Paz University Hospital. All participants provided written informed consent in accordance with the ethical guidelines of the 1975 Declaration of Helsinki and the Committee for Human Subjects of La Paz University Hospital (HULP: PI-3521) and the Clinical Research Ethics Committees of the Hospitals of Madrid (17.10.1125-GHM). Peripheral blood mononuclear cells (PBMCs) from healthy volunteers’ blood were isolated by Ficoll-Plus gradient (GE Healthcare Bio-Sciences, Pittsburgh, PA, USA). Macrophages were obtained from PBMCs by a previously described adherence-positive selection protocol [31] and cultured in 6-well plates (Roche) with DMEM (Gibco-BRL Life Technologies) supplemented with 10% FBS (Gibco-BRL Life Technologies) and 0.01% penicillin/streptomycin (Thermofisher Scientific) at 37 °C in a humidified atmosphere with 5% CO_2_.

After 15 days of macrophage differentiation, cells were either stimulated with IL-4 (10 ng/mL, PeproTech, PeproTech Ltd., London UK) or co-cultured with J82, UMUC1 or 5637 BC cell lines in a 1:5 tumor:macrophage ratio for 24 h. After that, cells were treated or not with 1 mM PBA for an additional 24 h. All the reagents used for the cell culture were endotoxin-free, as assayed with the Limulus amebocyte lysate test (Cambrex, Charles City, IA, USA).

### 2.7. Lymphocyte Polarization and Intracellular Cytokine Staining

PBMCs from healthy donors’ fresh blood were cultured in RPMI 1640 medium (Gibco) containing 10% of FBS and stimulated 50% (*v*/*v*) with supernatant from macrophages treated with 1 mM of SB and PBA. Stimulation was performed for 6 h at 37 °C 5% CO_2_ in the presence of Golgi-Plug containing Brefeldin A (BD Biosciences, San Jose, CA, USA) and Golgi-Stop containing Monensin (BD Biosciences) added after 1 h of the stimulation according to the manufacturer’s instructions. After that, PBMCs were washed and stained with the Live/Dead fixable Blue (Invitrogen) and surface markers (listed in Appendix A) for 30 min at room temperature, twice washed, fixed and permeabilized using the eBioscience™ Transcription Factor Fixation/Permeabilization (Invitrogen) according to the manufacturer’s instructions. Subsequently, the fixed and permeabilized PBMCs were staining using fluorochrome-conjugated antibodies against intracellular makers listed in Appendix A. Labeled cells were acquired on a Cytek Aurora Spectral Cytometer (Cytek Biosciences, Fremont, CA, USA). Data were analyzed using FlowJo (TreeStar, Ashland, OR, USA) v10.6.2 software.

### 2.8. Vital Colorant Assays

Macrophages were stained with DiD and tumor cells with DiO following the manufacturer’s instructions (Vybrant Multicolor Cell-Labeling kit, Fisher). Stained macrophages were co-cultured for two days with three different BC cell lines (J82, UMUC1, 5637). Cells were cultured following a 1:5 ratio tumor:macrophage. Co-cultured stained cells were fixed with PFA 4% and mounted with mowiol containing DAPI (Vector Laboratories, Burlingame, CA, USA). Images were obtained with Leica-DMI400D (LEICA, Wetzlar, Germany) at different magnifications.

### 2.9. Cell Sorting and Flow Cytometry

For cell sorting, co-cultured cells were stained in PBS (Gibco-BRL Life Technologies) with APC anti-human EPCAM (BioLegend, Cat: 324208San Diego, CA, USA) and FITC anti-human CD11b (Beckman Coulter GmbH, Cat: IM0530, Krefeld, Germany) in the dark for 30 min at 4 °C at the recommended manufacturer’s concentration for each antibody. After that, cells were washed and resuspended in DMEM (Gibco-BRL Life Technologies) with 2% FBS (Gibco-BRL Life Technologies) media for cell sorting in BD FACS InfluxTM Cell sorter (BD Biosciences). For PD-L1 flow cytometry analysis, cells were labeled with the above-mentioned antibodies plus PE anti-human PD-L1 (BD Biosciences, Cat: 557924) for 30 min at 4 °C in PBS (Gibco-BRL Life Technologies). After that, cells were washed and analyzed in a BD FACSCalibur flow cytometer (BD Biosciences). The obtained data were analyzed with FlowJo 9.1v software (Tree Star).

### 2.10. Soluble Cytokine Quantification

The cytokine levels in culture supernatants were measured using the bead-based multiplex assay, LEGENDplex Human Inflammation Panel 1 (12-plex: IL-1β, IL-2, IL-4, IFN-γ, TNF-α, MCP-1 (CCL2), CXCL10, IL-6, IL-8 (CXCL8), IL-10, IL-12p70 and IL-17A) (BioLegend) according to the manufacturer’s instructions. Samples were analyzed on a FACSCalibur flow cytometer (BD Biosciences). Data were analyzed using LEGENDplex (BioLegend) v.8 software.

### 2.11. Statistical Analysis

Appropriately, t-student, two-way ANOVA followed by Tukey analysis or paired *t*-tests were performed. We also indicated sensitivity and the 95% confidence intervals (95% CI). Significance was set (* *p* ≤ 0.05, ** *p* ≤ 0.01, *** *p* ≤ 0.001, **** *p* ≤ 0.0001) using Prism 9.0 software (GraphPad).

## 3. Results

### 3.1. PBA and SB Induce Cell Cycle Arrest in BC Cell Lines and Impacts Their Immune Profile

To evaluate the effect of both butyrate derivatives, three BC cell lines expressing high levels of Class I and Class II HDACs (5637, UMUC1 and J82) were treated with increasing concentrations of PBA and SB. In cell cycle analysis both PBA and SB augmented the number of cells in G1 and sub-G1 phase (Figure 1A) in all the assayed cell lines. The sensitivity to both compounds was determined by IC50 calculation. For both compounds, 5637 and UMUC1 cell lines showed IC50 values between 4 and 8 mM, whereas J82 showed less sensitivity with IC50 of 13.8 mM for PBA and over 23 mM for SB (Appendix A). The effect of butyrate derivatives as HDACis was demonstrated by the increase in histone H3 and H4 total acetylation and for the H3K9ac-specific mark (Figure 1B). As expected, both PBA and SB decreased Cyclin D as well as increased p21 expression (Appendix A). Levels of p27 were also increased in UMUC1 and J82, but not in 5637 cells (Appendix A). Additionally, the effect of both butyrate derivatives was evaluated by gene expression analysis, confirming the observed protein increase in *CDKN1A* (p21) and *CDKN1B* (p27) at the mRNA level, which was further reinforced by a decrease in E2F1 and MYC (Figure 1C).

To evaluate the effect of both compounds on genes/proteins previously demonstrated to be important in the tumor–macrophage interaction [12], we also evaluated the gene expression of *CD274* (*Pdcd1*, PD-L1), *TLR4*, *BMPR2* and miR-21 by RT-qPCR on the treated cell lines. Expression levels of *CD274* were increased upon treatment with both compounds for UMUC1 and with PBA for 5637 cell lines, although no significant changes were observed for J82 cells (Figure 1D). Similar results were obtained for *TLR4*, whose expression was increased in UMUC1 after both treatments and in 5637 cells only after SB (Figure 1D). *BMPR2* and miR-21 expression were not affected by PBA and SB treatment in any of the evaluated cell lines (Figure 1D).

### 3.2. Butyrate Derivatives Increase PD-L1 Expression, Induce Non-Classical Phenotype on Monocytes and Reduce M2 Polarization in Human Macrophages

To determine the effect of both butyrate compounds on human monocytes, we purified peripheral blood monocytes from human donors and treated them with three doses of PBA and SB (0.5, 2 and 4 mM). Flow cytometry analysis revealed that both treatments reduced the CD14 density on the monocyte cell surface (Figure 2A) but only SB reduced the percentage of CD14-positive cells (Figure 2B). We did not find any relevant changes in HLA-DR expression after the abovementioned treatments; however, a significant increase in the PD-L1 molecule was observed when the monocytes were treated with PBA. The treatment with SB was less effective compared to PBA (Figure 2C,D).

Since HDACis have been reported to induce changes in cytokine profiles, we also measured some soluble factors known to be important in the TME–tumor interaction. As shown in Figure 2E, IL-1β expression was increased and IL-8, IL-6, IL-10 and TNFα levels were decreased upon PBA treatment. In the case of SB these changes were not so clear.

Once it was demonstrated that butyrate compounds have undeniable effects on innate immune cells and inflammatory cytokine production, we decided to check if PBA affects the M2 polarization of human macrophages, since is it known that M2 macrophages play a major role in BC [12]. We found that PBA reduced the IL-4-induced M2 polarization of human macrophages as shown by decreased IL-10 production, reduced CD163 expression, and increased HLA-DR expression on M2 macrophages (Figure 2F).

To further explore the effect of these compounds in the immune tumor microenvironment, we evaluated the culture supernatant of SB- or PBA-treated macrophages (Appendix A) on naïve PBMCs’ production of cytokines (with the addition of protein transport inhibitors Brefeldin A and Monensin to enhance their detection). We analyzed the expression of different intracellular cytokines (IL-2, IFNγ, TNFα, IL-4 and IL-17), the transcription factor FoxP3 and the CD25 surface marker to study Th1, Th2, Th17 and Tregs responses in CD4^+^ T cells. We observed an increase in IL17 production. (Suplementary Appendix A) revealing that although they do not affect the cytotoxic CD8^+^ nor the Treg, PBA significantly affect the polarization of helper CD4^+^ into effector Th17 lymphocytes reinforcing its contribution to an inflammatory microenvironment.

### 3.3. Macrophage–BC Cell Co-Cultures Generate a THC Population with Enhanced PD-L1 and Pluripotency Genes Expression

Using the three BC cell lines previously shown to induce M2-like macrophage polarization (UMUC1, 5637 and J82) [12], we performed co-cultures with monocyte-derived macrophages isolated from healthy donors [32]. After 48 h of co-culture following the gating strategy from Figure 3A, cells were separated based on the surface expression of CD11b^+^ (macrophages), EPCAM^+^ (tumor cells) or both surface markers EPCAM^+^ CD11b^+^ (hybrid population). In all the assayed cell lines, we obtained a population close to 2% of total cells that represented hybrids expressing both CD11b and EPCAM on their surface (Figure 3B). Curiously, the J82 cell line showed low EPCAM expression but was the cell line with highest hybrid population counts. To visualize these hybrid cells, we stained the co-cultures with vital colorant DiD for macrophages and DiO for BC cells. Double-positive cells defined as DiD^+^DiO^+^ were rapidly detected by confocal microscopy (Figure 3C).

We and others have previously reported that these hybrid cells commonly form by fusion between cancer stem cells (CSCs) and monocytes/macrophages, resulting in a new cellular entity with immunomodulatory ability [6,33]. To characterize this hybrid population, we evaluated surface PD-L1 levels as well as the expression of pluripotency associated genes such as *MYC*, *SOX2* and *NANOG*. As shown in Figure 4A, PD-L1 expression was higher in the hybrids formed from UMUC1 and 5637 cells in comparison with the EPCAM^+^ or CD11b^+^ cells. No differences were observed for the J82 cell line. Regarding the pluripotency genes, hybrid populations were enriched in *SOX2* and *NANOG* expression when compared with the EPCAM^+^ or CD11b^+^ cells (Figure 4A). No differences were found for *MYC* gene expression.

Next, to study the macrophage–tumor cell interaction, we evaluated the cytokine expression profiles characteristic for M1 or M2 macrophage in co-culture supernatants. In comparison to cultured macrophages or BC cell lines, co-cultures showed increased levels of IL-10, IL-4, TNFα and IFN-γ according to the expected alternative M2 polarization. However, an increase in IL-1β, IL-6, CXCL-10, CCL-2 and TNFα was also observed in the co-cultured condition (Figure 4B).

### 3.4. THC Population Is More Sensitive to PBA Treatment Than Parental Cells

As PBA was more efficient than SB, we chose to evaluate the effect of PBA in a co-culture setting. Mature macrophages from three different healthy donors were co-cultured with each of the three cell lines for 24 h and subsequently treated with PBA 1mM for 24 h. As observed for the treatment of individual cell populations, PBA addition decreased *MYC*, *NANOG*, *SOX2*, *IL10* and *BMP4* expression and increased the surface expression of PD-L1 (Figure 5A). In co-culture supernatants, we observed a significantly decreased production of IL-10 and TNFα upon PBA treatment and an increase in IL-1β in all cell lines, as well as a decrease in IL4 and IFNg for 5637 (Figure 5B). No changes were observed for CXCL10, CCL2, Il6, IL8 and TGFβ (Appendix A).

The cells were then sorted into CD11b^+^, EPCAM^+^ and double-positive populations. The analysis of these purified cell populations demonstrated that THC showed a similar decrease in IL10 to the CD11b population for UMUC1 and J82 cell lines and a similar reduction to the EPCAM-sorted population for BMP4 gene expression (Figure 6A). Although hybrid formation during co-cultures was not reduced after treatment, the expression of genes related to pluripotency, such as *SOX2*, *NANOG* and *MYC*, was significantly reduced by PBA treatment (Figure 6B). In addition, the expression of two miRNAs involved in tumor progression and metastasis, miR-302 and miR-21, was evaluated. We found no differences in the treated parental tumor cells and macrophages. However, the PBA treatment significantly decreased miR-302 and miR-21 expression levels in the hybrid cell population (Figure 6C). These data indicate that, while PBA does not affect THC formation, it dampens their pluripotency potential.

## 4. Discussion

Histone-modifying enzymes that catalyze reversible lysine acetylation are major players in the epigenetic modulation of gene expression and are recognized targets in anticancer therapies [34,35]. The exact mechanism of action of PBA and SB as HDACi is not fully understood and has been shown to be dependent on the cell type. PBA has been shown to display the activity of a chemical chaperone at high concentrations and to possess the ability of inhibiting HDACs Class I and Class II [29]. PBA is characterized by good bioavailability in vivo and because of its low cytotoxicity an interesting area of investigation concerning its utility in oncology research has been opened [36,37]. Due to their growth-inhibiting and apoptosis-inducing activity, phenyl and sodium butyrate, alone or in combination with other anti-cancer drugs, have been evaluated for the treatment of several malignant tumors [38,39]. In addition to its anti-neoplastic effect, butyrate exerts a functional effect over the TME, particularly over macrophage polarization [26]. Zheng et al. [40] demonstrated that TAMs with an M2-like phenotype treated with pan-HDACis or HDAC2 siRNAs were re-polarized to M1-like cells with a decrease in IL-10 and CD206 expression and an increase in TNFα and IL-8 levels. As previously reported for BC cells, as well as for other tumor types, we observed a cell cycle arrest in G1 and increase in Sub-G1 phase in BC cells treated with PBA and SB. This phenotype is associated with increased expression of p21, a decrease in H3 and H4 acetylation and an induced expression of immune checkpoint molecules such as PD-L1. On the other hand, with respect to their effects on monocytes, we found that both PBA and SB induced the secretion of IL-1β and IL-8 as well as a decrease in CD14 expression, together with an increase in PD-L1 surface expression. Our results indicate that butyrate derivatives could be of benefit for the management of BC, affecting not only tumor cells but also the TME.

Cancer cell–macrophage interactions in the BC microenvironment have consistently been the focus of recent research due to their important effect on tumor progression and metastasis as well as their relation to therapeutic responses [12,41,42]. TAMs exhibiting an M2-like phenotype induce the cancer-cell-mediated secretion of tumor growth factors, cytokines such as IL-10, and proteases, which promote angiogenesis, connective tissue breakdown and tumor-cell proliferation [43,44]. Thus, the functional reprogramming of TAMs into a pro-inflammatory (M1-like) phenotype, based on their cellular plasticity, is a key area of the BC–TME modulation strategy with the main objectives of reducing the pro-tumorigenic effects of TAMs and to favor further anti-cancer strategies [45,46]. A high TAM density is associated with a poor prognosis in BC [47,48] and surface expression of CD163 correlates with tumor progression and worsened prognosis [48,49]. CD163 expression has been largely considered to be present only on the surface of the monocyte/macrophage lineage and has just recently been reported to be also expressed in some tumor cells characterized by an advanced histological grade, a higher occurrence of distant metastasis and reduced patient survival [50,51]. Our data in both BC cells/macrophage co-cultures and isolated macrophages strongly suggest PBA is a suppressor of the macrophage M2 polarization.

Maniecki et al. [11] demonstrated that after BC cell line–macrophage co-culture, some tumor cells expressed the CD163 surface marker, and Saed et al. [52] observed that epithelial ovarian cancer cells and tissue also expressed macrophage markers such as CD11b. In our macrophage–BC cell line co-culture experiments, we also obtained a THC population that showed increased pluripotency gene expression associated with CSCs. Huysentruyt et al. [53] reported metastatic tumor cells that expressed multiple properties of macrophages, including *ITGAM* (CD11b) gene expression in an in vivo mouse model. Recently, Gast et al. [33] reported that the fusion of cancer cells with macrophages results in cells with increased metastatic behavior. These fused hybrids were detectable in cell cultures as well as in tumor-bearing mice. Additionally, their numbers in the peripheral blood of human cancer patients act as a good indicator of metastatic potential. Taken together, targeting these cells could offer a new strategy to fight tumor metastasis either by eliminating these cells, by diminishing their formation or by reducing their metastatic capabilities.

Even though the total number of THC was not affected by PBA or SB, we observed that the CSC features of these cells were reduced by these treatments in macrophage–BC cell line co-cultures, which is indicated by a significant decrease in pluripotency markers such as *MYC*, *NANOG* and *SOX2* as well as miRNAs associated with cancer stemness such as miR-21 and miR-302. This effect was observed in sensitive BC cell lines (UMUC1 and 5637) as well as in J82 (shown to be resistant to butyrate treatment), indicating that the reduction in pluripotency markers is not associated with growth inhibition. Remarkably the increased pluripotency and the expression of some of these miRNAs is associated with increased metastatic behavior in other systems [54,55]. This potential reduction in metastatic capacity by PBA treatment will be the subject of future studies. In addition to the reduction in pluripotency gene expression, the hybrid cell population particularly showed an increased PD-L1 surface expression upon treatment. Despite the immunosuppressive role of PD-L1, its expression could be prompted by a more inflammatory microenvironment. Importantly, a recent publication demonstrated the efficacy of combining HDAC inhibition with PD-1 blockade for the treatment of melanoma in a murine model. Mice receiving combination therapy had a slower tumor progression and increased survival compared to control and single-agent treatments [56]. Furthermore, since the increased expression of PD-L1 in tumor cells and the stromal compartment is an indicator of a better response rate to immune checkpoint blockade therapies, the observed induction of PD-L1 could provide a potential strategy to augment the benefit of immunotherapies through the conditioning of the TME with HDACi [57,58].

## 5. Conclusions

A holistic understanding of the effect of anticancer therapies on not only the tumor, but also its microenvironment is, more than ever, a must. The identification of hybrid cells between leucocytes and tumor stem cells as well as their importance in tumor cell propagation opens a new avenue of targets to limit tumor spread and metastasis. Since previous experiments by Lopez-Collazo and collaborators [6] have shown that macrophage-fusing epithelial cells display a more mesenchymal phenotype and Gast et al. [33] have demonstrated the hybrids’ increased metastatic spread, our findings support that treatment with PBA could reduce the hybrids’ metastatic capability in tumors sensitive and resistant to HDACis.

## Figures and Tables

**Figure 1 cancers-14-00287-f001:**
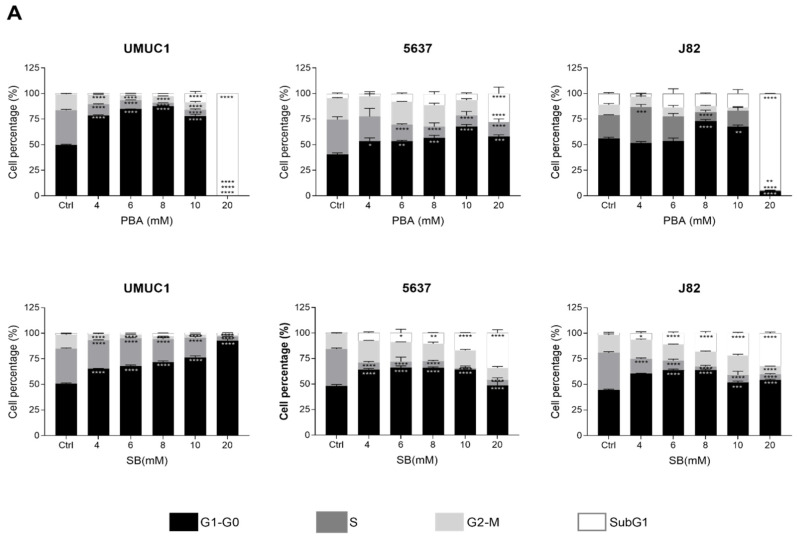
Effect of PBA and SB treatment on BC cell lines UMUC1, 5637 and J82. (**A**) Cell cycle phase percentage of UMUC1, J82 and 5637 BC cells without treatment (Ctrl) and after treatment with increasing concentrations of PBA and SB. Each bar corresponds to UMUC1, J82 and 5637 BC cells without treatment (Ctrl) and after treatment with PBA and SB as shown. (**B**) Immunoblot showing the expression of histone 3 and 4 acetylation in UMUC1, J82 and 5637 BC cells without treatment (Ctrl) and after treatment with PBA and SB. Quantification levels of all marks in the same cells are also shown. Total H3 and H4 were used for loading normalization. (**C**) Gene expression quantification by qPCR of cell-cycle-related genes p21 (*CDKN1A*), p27 (*CDKN1B*), *MYC* and *E2F1*. (**D**) Gene expression of molecules related to tumour–cell interaction such as CD274, TLR4, BMPR2 and miR-21. Expression of coding genes was normalized with respect to *GUSB* and miR-21 with respect to RNU6. Significance is depicted as follows: * *p* ≤ 0.05, ** *p* ≤ 0.01, *** *p* ≤ 0.001, **** *p* ≤ 0.0001.

**Figure 2 cancers-14-00287-f002:**
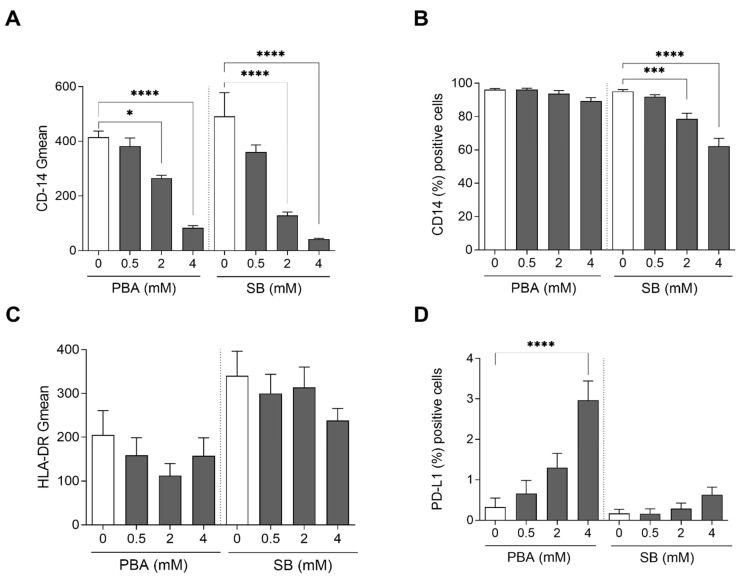
PBA and SB modulate phenotype and polarization of human monocytes and macrophages. CD14 (**A**) and HLA DR (**C**) geometric mean (Gmean) and percentage of CD14- (**B**) and PD-L1 (**D**)-positive human monocyte cells after treatment with PBA and SB at different concentrations (0, 0.5, 2 and 4 mM). (**E**) Heatmap containing cytokine concentration after treatment with different concentrations of PBA and SB. (**F**) Monocyte-derived macrophages were treated with IL-4 for 24 h to induce M2 polarization. Later, cells were treated with 1 mM of PBA for another 24 h. Production of IL-10 (left panel) and expression of CD163 (central panel) and HLA-DR (left panel) are shown. Significance is depicted as follows: * *p* ≤ 0.05, *** *p* ≤ 0.001, **** *p* ≤ 0.0001.

**Figure 3 cancers-14-00287-f003:**
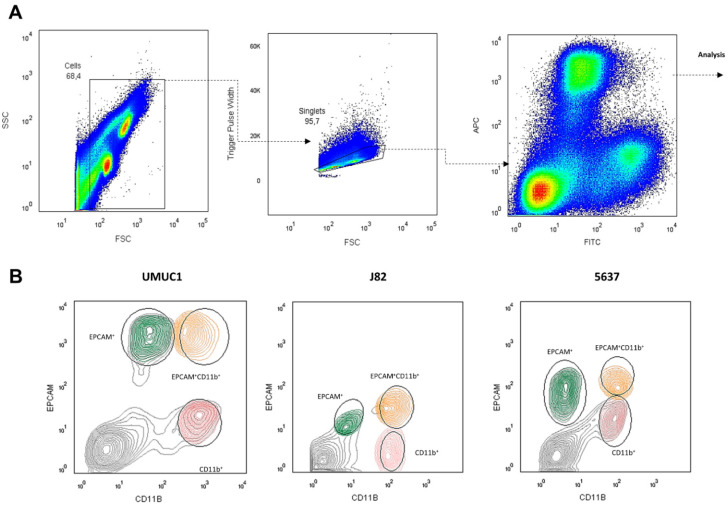
BC cell lines UMUC1, 5637 and J82 co-cultured with macrophages give rise to a hybrid population. (**A**) Gating strategy used for cell sorter. CD11b and EPCAM were used as cell surface markers for macrophages and tumor cells, respectively. The hybrid cell population is represented by CD11b^+^EPCAM^+^ cells. (**B**) Representative FACS analysis (*n* = 3; CD11b+ and EPCAM+) for THC characterization assays. (**C**) Representative confocal images of double-positive events with vital colorants DIO (green, BC cells) DID (red, macrophages) and DAPI for nuclei (blue), after 5 min. Single cultures of macrophages and BC cells were included as a control. Arrows indicates THC. Scale bars, 100 μm.

**Figure 4 cancers-14-00287-f004:**
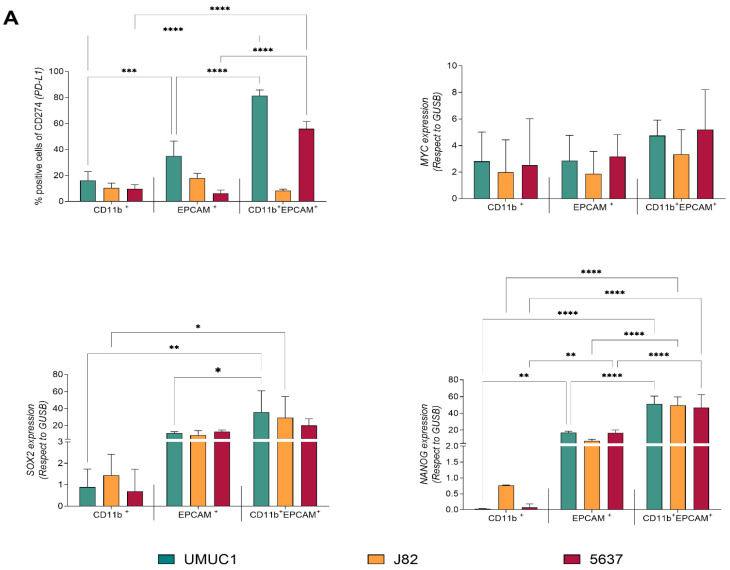
THC showed an enrichment in pluripotency factors compared with parental cells. (**A**) PD-L1 surface expression and *MYC*, *SOX2* and *NANOG* gene expression in the assayed cell populations. PD-L1 surface expression is represented as the percentage of cells expressing PD-L1 on their surface. *MYC*, *SOX2* and *NANOG* gene expression is shown relative to housekeeping *GUSB* gene expression. (**B**) Evaluation of cytokine expression profiles characteristic for M1 or M2 macrophage polarization in co-culture supernatants. IL-4, IL-10, IL-1β, TNFα and IFN-γ cytokine expression profiles are shown for single culture and co-culture supernatants. Supernatants of macrophages and BC cells single cultures were included as a control. Significance is depicted as follows: * *p* ≤ 0.05, ** *p* ≤ 0.01, *** *p* ≤ 0.001, **** *p* ≤ 0.0001.

**Figure 5 cancers-14-00287-f005:**
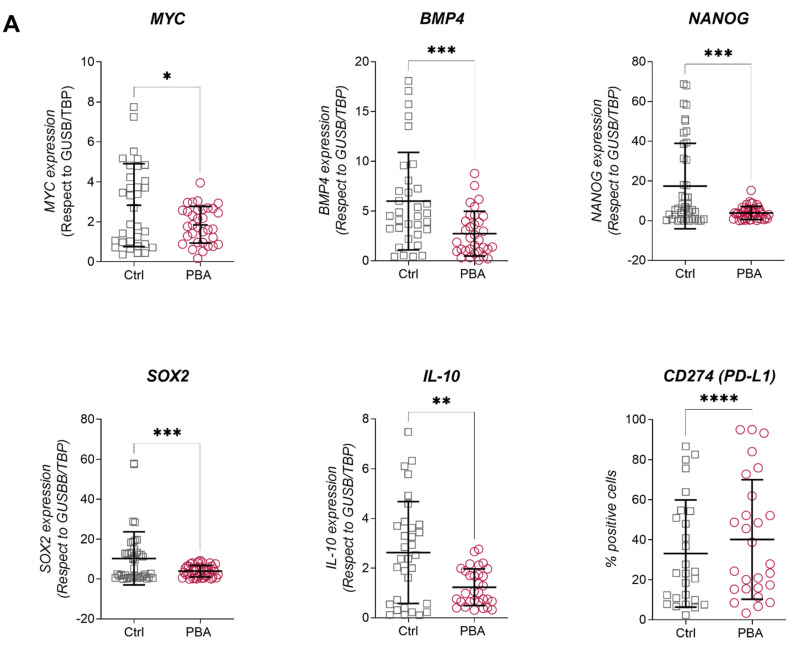
PBA treatment decreased pluripotency factors and affected cytokine production. (**A**) *MYC*, *IL-10*, *BMP4* and PD-L1 expression levels in all populations of three BC cell lines before and after treatment with PBA. Expression of coding genes was normalized with respect to *GUSB*. PD-L1 surface expression is represented as the percentage of cells expressing PD-L1 on their surface. (**B**) IL-10, IL-4, IL-1β, CXCL-10, TNFα, CCL-2, IL-6 and IL-8 levels in cell culture supernatant after treatment with PBA before and after co-culture with macrophages. Significance is depicted as follows: * *p* ≤ 0.05, ** *p* ≤ 0.01, *** *p* ≤ 0.001, **** *p* ≤ 0.0001.

**Figure 6 cancers-14-00287-f006:**
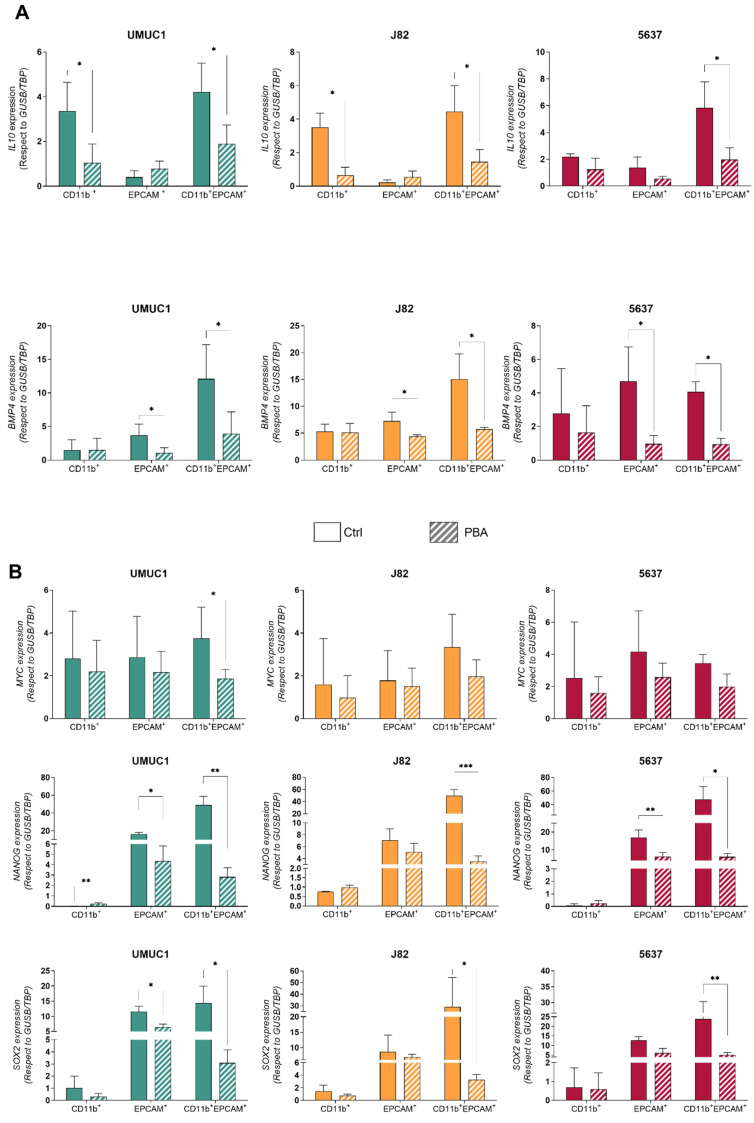
THC have enhanced sensitivity to PBA compared with their parental cells. (**A**) *IL-10* and *BMP4* expression levels in CD11b^+^, EPCAM^+^ and CD11b^+^EPCAM^+^ cells with respect to *GUSB/TBP* in UMUC1, J82 and 5637 before and after treatment with PBA. (**B**) *MYC*, *NANOG* and *SOX2* expression levels in CD11b^+^, EPCAM^+^ and CD11b^+^EPCAM^+^ cells with respect to *GUSB/TBP* in UMUC1, J82 and 5637 before and after treatment with PBA. (**C**) Expression levels of miR-21 and miR-302 normalized to *RNU6B* in CD11b^+^, EPCAM^+^ and CD11b^+^EPCAM^+^ cells in UMUC1, J82 and 5637 cell lines. The points represent experimental replications. Significance is depicted as follows: * *p* ≤ 0.05, ** *p* ≤ 0.01, *** *p* ≤ 0.001.

## Data Availability

All data supporting reported results can be found in figures and provided Appendix A.

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
