# Peer review of "Toward Tumor Fight and Tumor Microenvironment Remodeling: PBA Induces Cell Cycle Arrest and Reduces Tumor Hybrid Cells’ Pluripotency in Bladder Cancer"

_cancers, 2022, doi:10.3390/cancers14020287_

Round 1

Reviewer 1 Report

The authors should check the references. I noticed some of them are not in the correct format.

Reviewer 2 Report

The authors have made valiant effort to revise whatever they could and the manuscript has good concept and experimental design that will help contribute knowledge to the scientific community. 

Reviewer 3 Report

I appreciate the changes the authors did to improve the manuscript.

This manuscript is a resubmission of an earlier submission. The following is a list of the peer review reports and author responses from that submission.

Round 1

Reviewer 1 Report

In this work, the authors analyzed the effects of butyrate derivatives PBA and SB on some bladder cancer cell lines and on some subsets of immune cells. They observed that butyrate derivatives induced a cell cycle arrest of cancer cell lines and modulated the expression of markers of monocytes/macrophages immune subsets. Moreover, they observed that co-culture experiments (cancer cell lines and macrophages)  generated a hybrid population, characterized by an increased expression of specific genes associated to pluripotency, including NANOG and SOX2 and PDL1. Finally, they observed this hybrid population is susceptible to PBA treatment that decreased the expression of pluripotency-linked genes and cytokines encoding genes, including IL10 and TNF-alpha.

The work is really interesting however, in my opinion, the key message of the authors is not fully clear. What is the effect of macrophages, treated with butyrate derivatives, on tumor growth? Is it possible to translate the experimental model shown by the authors to a murine model in which it is possible to monitor the tumor growth and the composition of immune subsets within the tumor microenvironment? I believe that authors could evaluate the effect of macrophages-treated with PBA or SBA (or cytokines produced by macrophage) on T-cells response CD4 (Th1, Th2, Th17, Treg) or CD8, analyzing the expression of specific markers by realtime PCR (As reference for this simple experiment, the authors could use PMID: 34604232)

Author Response

We really appreciated the comments and suggestions from this reviewer. We are very grateful for his positive feedback on our study. We have taken into account and tried to clarify and complete the points indicated by the reviewer, in the manuscript and below.

Please see the attachment for point by point response to the reviewer comments.

Reviewer 2 Report

Overall, the concept of the paper was good but still some missing links/work needs to be completed to complete their story. Below are my suggestions. Thank you

  • In simple summary state that BC is bladder summary
  • Phenylbutyrate acid (PBA) is a HDAC inhibitor but which HDACs does it inhibit?
  • Line 53 “This THCs” should be These THCs
  • In Fig 1A it would be good to include normal bladder cells you can use as controls?
  • Fig 1B: it would be good include where in histone 3 and 4 acetylations occur.
  • Do these BC cells have HDACs expressed? Need to show as proof-of concept since PBA is HDAC inhibitor
  • Fig 1C hard to read. The x-axis and y-axis are hard to read
  • Line 306 THC has previously defined so no need to define it here
  • For Fig 4A-4B: labels are blurry. Resolution is not good
  • Fig 5B is blurry
  • Fig 6 is blurry
  • What is the mechanism-of-action by which PBA results in increased PD-L1 expression and reduced the pluripotency molecular markers in THCs? Can even elude details in discussion.
  • Does PBA have a similar effect in vivo in BC Patient-derived xenografts? Need to discuss
  • Are butyrate derivatives currently used in cancers for treatment? If so, then are the doses of PBA and SB being used here clinically-relevant?
  • Need bit more discussion or mechanism studies on how PBA impacts histone deacetylation.
  • It may also be good to use another HDACi (belinostat, panobinostat or vorinostat) along side with PBA to see if similar results are obtained.

Author Response

We really appreciated the comments and suggestions from this reviewer. We are very grateful for his positive feedback on our study. We have taken into account and tried to clarify and complete the points indicated by the reviewer, in the manuscript and below.

Please see the attachment for point by point response to the reviewer comments.

  • In simple summary state that BC is bladder summary

It is now stated that BC is bladder cancer.

  • Phenylbutyrate acid (PBA) is a HDAC inhibitor but which HDACs does it inhibit?

We thank the reviewer for this comment. PBA in an aromatic short fatty acid known for inducing differentiation, cell cycle arrest, and apoptosis in various cancer cells. However, the effects of PBA seem to be mostly cell-type-specific. PBA (and SB) has been demonstrated to be a pan-HDAC inhibitor acting over class I and class II HDACs. Among various activities of PBA, it has been demonstrated to be the reversible inhibitor of class I and II HDACs (World J Gastroenterol. 2013; 19(8):1173-1181) PBA mode of action in cancer cells has been attributed to reduced proliferation (Chin Med J (Engl) 2008 and Cytokine 1995 Jul;7(5):449-56), enhanced differentiation (Cytokine 1995 Jul;7(5):449-56 and Int J Oncol. 2008;32(4):821–827), increased apoptosis (Int J Oncol. 2008;32(4):821–827; J of Cellular Biochemistry 2005, 93(4) 819-829 and World J Gastroentero 2012 Jan 7;18(1):79-83), and cell cycle arrest (World J Gastroentero 2012 Jan 7;18(1):79-83 and Int J Radiat Oncol Biol Phys 2007 Sep 1;69(1):214-20). However, the molecular pathways underlying these processes are still not well understood. This point has been addressed in the background section of the manuscript.

  • Line 53 “This THCs” should be These THCs

We thank the reviewer for this comment. It has been corrected in the present version.

  • In Fig 1A it would be good to include normal bladder cells you can use as controls?

We accept the point from the reviewer, however we have bought non transformed urothelial cell from different commercial sources, but they do not grow in culture conditions, so it has been impossible to do that. In this experiment the control is set up with the cell lines without treatment since the objective is to evaluate the effect of the compounds on the tumor cells.

  • Fig 1B: it would be good include where in histone 3 and 4 acetylations occur

We thank the reviewer for this suggestion. Accordingly, we evaluated specific H3K9 lysine residue in addition to the total H3 and H4 known to be acetylated in BC cell lines and effectively demonstrated that both treatments increase acetylation as expected. This result is now included in the manuscript as new Figure 1B.

  • Do these BC cells have HDACs expressed? Need to show as proof-of concept since PBA is HDAC inhibitor

We thank the reviewer for this suggestion. To specifically evaluate the HDAC expression as well as the overall PBA and SB treatment effect upon the assayed BC cell lines we have evaluated histone expression in the BC cell lines (see attached figure). We observed higher expression of HDAC3, 4, 5, 6, 9 and 11, lower expression levels of HDAC 1, 2 and 8. HDACs 7 and 10 was not possible due to technical problems. The expression levels of these molecules were not affected by the treatment with PBA or SB. These results are mentioned in the manuscript but as results not shown. We also observed that the expression levels were not affected by the treatment with PBA or SB. To avoid possible complexity an maintain the length of the manuscript into admissible limits, these results are mentioned in the manuscript as results not shown..

  • Fig 1C hard to read. The x-axis and y-axis are hard to read

 We agreed with the reviewer comment. We have now split the figure and enlarged the axis legend. We hope it is clearly visible now

  • Line 306 THC has previously defined so no need to define it here

The reviewer is right. It has been corrected in the present version.

  • For Fig 4A-4B: labels are blurry. Resolution is not good.

The reviewer is right. It has been corrected in the present version.

  • Fig 5B is blurry

The reviewer is right. It has been corrected in the present version.

  • Fig 6 is blurry

The reviewer is right. It has been corrected in the present version.

  • What is the mechanism-of-action by which PBA results in increased PD-L1

expression and reduced the pluripotency molecular markers in THCs? Can even elude details in discussion.

PBA displays potentially favorable effects on many pathologies including cancer, genetic metabolic syndromes, neuropathies, diabetes, hemoglobinopathies, and urea cycle disorders. The mechanisms by which PBA exerts these effects are different. Some of them relate to the regulation of gene expression, playing a role as a putative histone deacetylase inhibitor, while others, as chemical chaperone, contribute to rescue conformational abnormalities of proteins, and some others are attributed to its capacity of enabling excretion of toxic ammonia, acting as ammonia scavenger. Defining the exact mechanism of action or which of these effects is the one related to the increase of PDL1 at this point is speculative. However, in a recent paper (Oncogene volume 40, pages1836–1850 (2021)) the authors observed that the treatment with TSA (an HDACi) increases PDL1 expression in melanoma and lung cancer mouse model and that combination of TSA and anti-PDL1 treatment in this model significantly enhanced the durability of tumor reduction and prolonged survival of tumor-bearing mice, compared with the effect of either treatment alone. As we discussed on the present version this could be a precedent for a new combined therapy for treating BC.  

  • Does PBA have a similar effect in vivo in BC Patient-derived xenografts? Need to discuss PDX?

We agree with the reviewer regarding the relevance of analyzing the effects in vivo. This possibility is considered for future studies using different immunocompetent transgenic mouse models generated by our group, as PDX models lack functional immune system. Indeed, we have previously reported the generation of different immunocompetent mouse models of BC by genetic manipulation (Nat Med. 2019 Jul;25(7):1073-1081; Clin Cancer Res. 2019 Jan 1;25(1):390-402; Cancer Res. 2014 Nov 15;74(22):6565-6577). Nonetheless these models also have several caveats (regarding the time of appearance of tumors, asynchronous growth, different tumor types and different progression rates) that require further development. This refinement is on its way but would require more experimental work. Once we have solved these issues, we certainly would like to monitor the effects of different antitumor compounds (including epigenetic inhibitors) focusing not only on tumor cells but also on microenvironment cells (including different immune populations).  However, at present, we consider that this fall beyond the scope of the present manuscript.

  • Are butyrate derivatives currently used in cancers for treatment? If so, then are the doses of PBA and SB being used here clinically-relevant?

At present we have found 10 clinical trials analyzing this possibility in the ClinicalTrials.gov database, most in brain and colorectal tumors. or in patients with metastatic tumors that have not responded to previous treatments. It is difficult to be sure of the in vivo doses exactly corresponds with the in vitro concentrations used here, but the IC 50 reported in the literature for different tumor cells varied between 1 and 10 mM (as with many of the currently assayed HDACi such as Vorinostat or Panobiostat). Nonetheless, the possible analysis of antitumor (or anti metastatic dissemination) effects in vivo would require a prior detailed analysis to determine the correct doses and the possible secondary adverse events.

  • Need bit more discussion or mechanism studies on how PBA impacts histone deacetylation

We appreciated the reviewer comment. In the revised manuscript we have now included a more detailed discussion on the mechanisms of PBA on Histone acetylation, including a particular study in colon cancer showing that butyrate derivatives inhibited proliferation of stem cells through a Foxo3-dependent mechanism.

  • It may also be good to use another HDACi (belinostat, panobinostat or vorinostat) along side with PBA to see if similar results are obtained.

We agree with the reviewer. We have evaluated Vorinostat in the same three cell lines evaluated with PBA and SB. We enclosed here the obtained results. As summarized in the figure we observed that Vorinostat treatment also reduced the stemness factors and IL10 production by CD11b sorted cells as we previously observed with PBA and when analyzing the cytokine profiles we observed, similarly than with PBA, an increase of IL1b and a decrease of IL10. However, instead of an increase we obtained a reduction of PDL1. More experiment should be done in this regard to further dissect if the PDL1 increase is related to the other effects of PBA (such as ER stress or metabolic changes) contrary to what it have been reported by that published an PDL1 increase upon HDACi treatment on melanoma in vitro and in vivo (DOI: 10.1158/2326-6066.CIR-15-0077-T)). Since this experiment do not enter in the scope of the present study, we decided not include it in the manuscript.

Reviewer 3 Report

In the manuscript Rubio et al., authors describe the effects of butyrate derivatives on bladder cancer cell lines, monocytes and bladder cancer-macrophage hybrid cells. The results showed impaired M2 macrophage polarization and reduction of pluripotency markers in tumor hybrid cells. The research was well conducted, the article is well written and provides interesting data on compounds for potential immunotherapy against bladder cancer.

In line 344, authors wrote “THCs were most sensitive cells to PBA as reflected by changes in specific gene expression upon treatment (Figure 6A)”. In my view, except for IL10 expression in the 5637 cell line, the graphs of Figure 6A do not show clear differences among the hybrid cells and the parental cells upon treatment. Can the authors comment on/or clarify their statement?

Minor issues:

The PBA/SB concentration used in the experiment of figure 1B (Immunoblot) is missing.

Graph resolution of some figures are suboptimal. For instance Fig 1C, D.

In figure 3B, the EPCAM+ cluster of J82 cell line in the dot plot is not easily visible. Presumably, the gate is causing some transparency effect on the plot cluster or the green color chosen for the gate is too light. 

Author Response

(The authors gave the same response as above.)
